# Antimicrobial Food Packaging with Biodegradable Polymers and Bacteriocins

**DOI:** 10.3390/molecules26123735

**Published:** 2021-06-18

**Authors:** Małgorzata Gumienna, Barbara Górna

**Affiliations:** Laboratory of Fermentation and Biosynthesis, Department of Food Technology of Plant Origin, Poznań University of Life Sciences, Wojska Polskiego 31, 60-624 Poznań, Poland; barbara.gorna@up.poznan.pl

**Keywords:** active packaging, antimicrobial packaging, chitosan, bacteriocins, polylactide

## Abstract

Innovations in food and drink packaging result mainly from the needs and requirements of consumers, which are influenced by changing global trends. Antimicrobial and active packaging are at the forefront of current research and development for food packaging. One of the few natural polymers on the market with antimicrobial properties is biodegradable and biocompatible chitosan. It is formed as a result of chitin deacetylation. Due to these properties, the production of chitosan alone or a composite film based on chitosan is of great interest to scientists and industrialists from various fields. Chitosan films have the potential to be used as a packaging material to maintain the quality and microbiological safety of food. In addition, chitosan is widely used in antimicrobial films against a wide range of pathogenic and food spoilage microbes. Polylactic acid (PLA) is considered one of the most promising and environmentally friendly polymers due to its physical and chemical properties, including renewable, biodegradability, biocompatibility, and is considered safe (GRAS). There is great interest among scientists in the study of PLA as an alternative food packaging film with improved properties to increase its usability for food packaging applications. The aim of this review article is to draw attention to the existing possibilities of using various components in combination with chitosan, PLA, or bacteriocins to improve the properties of packaging in new food packaging technologies. Consequently, they can be a promising solution to improve the quality, delay the spoilage of packaged food, as well as increase the safety and shelf life of food.

## 1. Introduction

Active antimicrobial packaging serves to protect food products by actively modifying the internal environment and continuous interaction with food in a specified shelf life. Active packaging can be defined as a system that modifies the environment inside a food’s packaging, thus changing the condition of the packaged food system and its space to improve its quality by extending shelf life, improving sensory properties, and maintaining microbiological safety [1,2]. The growing popularity of active packaging results from consumers’ desire for high-quality, safe, and natural products [3]. In this technique, packaging materials actively interact with the food product. Some active packaging systems include O_2_ or CO_2_ scavengers, ethylene and moisture absorption systems, systems that emit CO_2_ or ethanol, and systems that release or contain antioxidants [3,4,5,6].

The traditional task of packaging is to protect against external factors and to provide the consumer with basic information about the product. Food quality and safety are important goals of the food industry. In this respect, food packaging ensures the quality and safety of food products until they reach consumers [7,8,9]. Active packaging increases shelf life and preserves product quality throughout the entire storage period. It can also be used for convenient food that is intended for quick preparation and consumption [10,11]. There has been an increase in recent years in the production of environmental biodegradable packaging from microbiologically synthesized natural polymers, such as polylactide (PLA) and chitosan, which are particularly useful in food packaging [10,12,13]. Combining them with added natural antioxidants allows to increase the antimicrobial effect of such films [13] and to inhibit the growth of food pathogenic microflora [14,15]. Modern food packaging technologies often include active packaging systems made of polymer-based materials. Through its interaction with the food product, this packaging can extend the shelf life of the product or improve its nutritional and sensory properties [7,9]. By interacting with the food surface or the free space inside the packaging, the antimicrobial packaging prevents or inhibits the growth of pathogenic microorganisms. The direct inclusion of antimicrobials in food tends to weaken their biological activity against microorganisms, due to their diffusion through the matrix. However, it is expected that the use of packaging materials containing antimicrobials will translate into a more effective means of preserving food than direct incorporation. Controlled diffusion of antimicrobial compounds from packaging materials onto the surface of a food product can delay or inhibit the initial growth of unwanted microorganisms on the surface and create a residual activity that remains during food storage and sale [7,9,16].

## 2. Antimicrobiological Packaging

Antimicrobial packaging is a form of active packaging that can extend the shelf life of the product, providing consumers with microbiological safety. It works to reduce, inhibit, or delay the growth of pathogenic microorganisms in packaged food and packaging materials [17].

To combat undesirable microorganisms on food surfaces, volatile and nonvolatile antimicrobials can be added into polymers, or a coating or adsorbent antimicrobial agent can be used on polymer surfaces [18]. Such a coating can be applied as a carrier for antimicrobial compounds or antioxidants to maintain high concentrations of preservatives on food surfaces [19]. 

The tactics in using antimicrobial packaging can be divided into two approaches [19]. The first represents the packaging materials where the antimicrobial surface is not in contact with the preserved food, and the active agents contained in them can migrate into the food. Such packaging is used for food that is wrapped in foil or under vacuum. The second approach is putting the antimicrobial agent inside the package but not in direct contact with the food [15].

Compounds that have been suggested for use in packaged antimicrobial foods are organic acids, enzymes (e.g., lysozyme), fungicides (benomyl, imazalil), and natural compounds with antimicrobial properties such as spices [19]. These compounds, apart from antimicrobial properties, have antioxidant properties. As natural food preservatives, nisin and lysozyme are safe for humans and can be used as additives to edible membranes [20,21]. One of the types of active packaging is antimicrobial packaging [22,23].

Antimicrobial packaging can limit or inhibit the development of undesirable microflora through the use of antimicrobial substances or through the antimicrobial properties of the polymer from which it is made [24,25]. There are five methods of producing antimicrobial packaging:Inserting a sachet (insert) containing an oxygen and moisture absorber, which reduces water activity and thus indirectly prevents the growth of microorganisms. The development of microflora is also inhibited by organic acids in such inserts [11,22,23,25,26];Including biostatic substances in the structure of the polymer (or paper), constituting the packaging by means of melting, extrusion, or the use of a solvent [11,22,26];Adsorption on the surface of the material of the antimicrobial compound [11,23];Creating covalent or ionic bonds between the polymer and the antimicrobial component, using a polymer with antimicrobial activity, such as chitosan [11,22,23,25].

### 2.1. Chitosan and Phenolic Compounds as Packaging Additives with Antimicrobial Properties

Chitin is the second most abundant polysaccharide on Earth after cellulose [27,28] and consists of three types of reactive functional groups, an amino group, and a primary and secondary hydroxyl group. Chitin is found in marine invertebrates, insects as a component of the exoskeleton, and in some fungi as a component of cell walls [28]. Marine invertebrates (crabs, shrimps, lobsters, and oysters) are consumed as a source of protein-rich marine food. However, the outer shells and other inedible parts of these crustaceans comprise about half of the body weight and are usually discarded as waste. This discarded waste is an excellent source of chitin [28].

The main source of commercial chitosan is chitin (Figure 1). It is found in green algae, fungal cell walls, cuticles of insects and arachnids, and in the exoskeleton of crustaceans [29,30,31,32].

Currently, most commercial chitosan is obtained from crustaceans (lobsters, shrimps and crabs), fungi, mollusks, arthropods, the cuticles of insects, and the scales of fish [30,31,32]. The main components of shellfish shells are chitin (15–40%), protein (20–40%), calcium, and magnesium carbonate (20–50%), along with other minor components such as astaxanthin, lipids, and other minerals [29,33].

However, fungi can also be a source of chitin. Fungal cell walls contain large amounts of chitin, which can be converted to chitosan by deacetylation reactions. For example, Bilbao-Sainz et al. [34] isolated chitosan from the stems of brown mushrooms and compared the physicochemical properties of fungal chitosan with high molecular weight (HW) and low molecular weight (LW) chitosan of animal origin. Fungal chitosan is a good membrane-forming material with properties similar to animal-derived chitosan membranes [34,35].

To obtain chitosan, several chemical processes should be used, such as decalcification, deproteinization, decolorization, and deacetylation (Figure 1) [33,36,37]. The demineralization of shells is usually carried out with a dilute HCl solution, but it is also possible to use other acids—HNO_3_, H_2_SO_4_, and CH_3_COOH [38]. The whole process takes place at room temperature [38]. The acid concentration and the time of shell treatment (demineralization) depend on the source of the chitin origin. Another process of deproteinization of the shells can be carried out with the use of diluted NaOH solution at a temperature of 65–100 °C for 0.5 to 72 h. Chitin deacetylation takes place at a temperature above 100 °C by hydrolysis of acetamide groups with concentrated NaOH or KOH (40–50%). This reaction is usually carried out under heterogeneous conditions, and the degree of acetylation (DA) of chitosan, defined as the proportion of acetylglucosamine units in the polymer, depends on the deacetylation conditions [33,38]. As reported in their research by Souza et al. [38], it is very difficult to completely deacetylate chitin without the use of specific procedures. The DA of chitosan generally ranges from 40 to 13%, which is of major importance in its solubility [33], and its molecular weight ranges from 2 × 10^5^ to 1 × 10^6^ Da [38].

An alternative method of obtaining chitin and producing chitosan from it are biological treatments that, unlike the methods using acids and bases, are safer for the environment [38]. Lactic acid-producing bacteria and bacterial proteases are used in the demineralization and deproteinization steps, respectively. Chitin deacetylation is carried out by enzymatic methods using chitin deacetylase. Despite the high quality of the final product and high environmental safety, this method requires a long processing time (several days) and so far is limited only to laboratory-scale tests [37,38].

In general, chitosan is prepared using conventional chemical and enzymatic methods, but ultrasound and microwave technology is added in some cases to improve the chitosan propriety through reduced its molecular weight [38,39].

Chitosan is a polysaccharide of *N*-acetyl-d-glucosamine and d-glucosamine units and it is mainly obtained by the partial deacetylation of chitin [6,28,32,40,41]. Chitosan is commercially obtained through partial deacetylation of chitin, leading to the formation of *N*-acetyl-glucosamine and d-glucosamine copolymers. Chitosan is a soluble form of chitin and has been used in various industrial applications including use in food preservation and packaging [6,9,28,38,42].

Chitosan (poly-(b-1/4)-2-amino-2-deoxy-d-glucopyranose) is the collective name for a group of partially and fully deacetylated chitin polymers. It functions as an antimicrobial polymer that can be used as an antimicrobial and polymer substrate simultaneously [28]. When used as ecological friendly packaging for food, chitosan films can extend shelf life and protect fresh food thanks to their good antimicrobial properties [32,43]. As a weak base, chitosan can dissolve in acidic solutions but remains insoluble in water. It can be produced by extrusion and pressing, which facilitates the use of antimicrobial properties in the form of films [6]. Chitosan has many excellent properties such as biodegradability, biocompatibility, bioactivity, nontoxicity, and polycationic nature [38,44]. It has found many applications alone or in combination with other natural polymers (such as starch, gelatin, alginates) in the food, pharmaceutical, textile, cosmetic, agricultural, and water treatment industries [45]. The antimicrobial activity of chitosan has been demonstrated against many bacteria, filamentous fungi, and yeast. Chitosan has a broad spectrum of activity and high activity against Gram-positive and Gram-negative bacteria, and has low toxicity to mammals [27,33,46,47].

The use of phenolic compounds and extracts in active packaging is of particular interest because these compounds have a strong antimicrobial effect in food systems and their consumption can contribute to improving human health [48].

Chitosan incorporated with extracts of propolis, mango leaf, and green tea resulted in active films with enhanced antioxidant activity [49,50]. 

Modifying chitosan using thermoplastic maize starch showed significantly higher antimicrobial activity against molds and yeasts [51]. The inclusion of ellagic acid in chitosan films improved its UV barrier and mechanical properties [50,52].

Recent studies on composite films with the addition of apricot kernel oil used to extend the shelf life of bread indicate the possibility of complete fungus growth inhibition [53]. What’s more, chitosan films have been modified with peanut peel extract, showed better lipid antioxidation for chicken meat and pork [54,55]. Most of these improved chitosan films possessed better mechanical strength and water vapor barrier ability than the pristine chitosan.

The compounds contained in the discussed extracts with strong antimicrobial properties are the already mentioned polyphenol compounds [48,49,50]. They belong to the natural antioxidants formed on the acetate-malonic pathway from carbohydrates. Their main source are plants, e.g., fruit and vegetables [49,56]. Depending on the structure, there are derivatives of hydroxybenzoic acid (gallic acid, ellagic acid); derivatives of hydroxycinnamic acid (*p*-coumaric, coffee, ferulic, and chloragenic acid); and stilbenes (e.g., resveratrol); tannins and flavonoids, including such subgroups such as flavonols (e.g., quercetin, campferol, myretin, morin, flavones (luteolin, apigenin); flavanones, flavanols, isoflavones (daidzein, genistein, glycitein); catechins, and anthocyanidins (cyanidine, delphinidin, malvidin, pelargonidine). Their common trait is the presence of phenolic groups in the molecules [56]. Gallic acid (GA) is a widely available phenolic acid with a strong antimicrobial effect [57]. It is commonly found in a number of fruits and vegetables, such as tea leaves, grapes, cherries, and longan seeds. From a structural point of view, GA is considered a trihydroxybenzoic acid [58,59]. It possesses antiallergic, anti-inflammatory, antimutagenic, and anticancer effects [59].

Gallic acid, extracted from *Caesalpinia mimosoides* Lamk. (Leguminosae) was active against *Salmonella typhi* and *Staphylococcus aureus* [57]. GA isolated from *Rosa chinensis* (Jacq.) flowers had significant antibacterial activity against pathogenic Vibrio bacteria [60].

Another quite interesting addition to food packaging also containing phenolic compounds can be grape seed extract (GFSE) [61]. GFSE contains phenolic compounds such as catechin, epicatechin, gallic acid, and procyanidins. It is a natural antimicrobial agent known to inhibit the growth of both Gram-positive and Gram-negative bacteria [61,62]. GFSE has strong antiseptic, antibacterial, fungicidal, and antiviral properties [61].

Tong et al. [63] combined GFSE with poly-ε-caprolactone, chitosan, and polyethylene to produce antimicrobial food packaging with very promising properties. The introduction of GFSE to biodegradable poly-ε-caprolactone increased the crystallinity of poly-ε-caprolactone and alginic acid. The smooth and homogeneous film is produced by compression molding; the film that is being formed also shows antibacterial activity against *P. aeruginosa* [63]. GFSE can be well dispersed in chitosan in various amounts (0.5, 1.0, and 1.5% *v/v*) without affecting the transparency of the film [61,63]. Increasing the addition of GFSE in the chitosan-based film improves its tensile strength and improves the elongation at break [63]. Preliminary evaluation of the antifungal activity has shown that chitosan-based composite films can retard fungal growth. The antimicrobial properties of polyethylene film obtained by co-extrusion or coating with GFSE solution were tested by applying it to ground beef at 3 °C [63]. The composite film showed increased antimicrobial activity against several microorganisms, including *E. coli* IFO 3301, *S. aureus* IFO 3060, and *B. subtilis* IFO 12113 [63].

Whereas Cha et al. [64] developed multi-component antimicrobials, including lysozyme, nisin, GFSE, and EDTA, that work alone or in combination. It turned out that GFSE-EDTA is a widely used antimicrobial agent against all indicator microorganisms, especially Gram-negative bacteria, in sodium alginate and K-carrageenan-based films [64].

Another interesting solution is the combination of antimicrobial chitosan, tea polyphenols, and silver nanoparticles (AgNPs) [65]. A novel one-pot method was used to create this nanocomposite film. The addition of tea polyphenols (TP) was used here, adding them to chitosan as a reducing agent for AgNPs, but above all, as a binding agent and a good antioxidant. Already the work carried out by Thomas et al. [66] has shown high antimicrobial effectiveness of films made of chitosan and silver nanoparticles against *Escherichia coli* and *Bacillus* bacteria. These films were obtained by synthesizing a silver particle with chitosan by reducing silver ions in an acid solution of AgNO_3_ [66].

On the other hand, the introduction of TP to chitosan, together with AgNPs, resulted in an improvement in the mechanical properties of the obtained composite, as well as in a higher antioxidant and antimicrobial activity [65]. Thus, the publicly available literature indicates the promising potential of polyphenolic compounds in the development of antimicrobial packaging materials to reduce the effects of pathogens and bacteria that contribute to food spoilage [49,50,63,65,66,67]. As packaging additives, they also improve the mechanical properties of the obtained chitosan-based composites [65,68].

Based on the data available in the literature, Table 1 presents some food products packaged in chitosan-based packaging as well as containing an antimicrobial agent in the formulation, which can enhance microbiological safety and possibly prolong food shelf life and quality.

### 2.2. Lactic Acid Fermentation Metabolites

Metabolites of lactic acid bacteria, such as bacteriocins [82], lactic acid, and other organic acids, can be used as antimicrobial substances in packaging that actively prevents the development of undesirable microorganisms in food products. Other compounds that can be used in this way are lysozyme and other enzymes, EDTA chelates, silver, and essential oils. Some substances are added to the packaging together to avail of synergistic effects [10,11,82,83].

The production of organic acids is undoubtedly a decisive factor in extending the shelf life and safety of the final product. Acidification is a widely used method of canning in the production of many types of food, such as fermented milk, vegetables, and sausages [84].

Lactic fermentation bacteria are not only tolerant of weak lipophilic acids, but also produce them as a byproduct of their metabolism. Some acids, such as acetic acid, are crucial for the metabolism of *Lactobacillus* bacteria, but inhibit *Bacillus* bacteria [85,86,87].

Organic acids show antimicrobial activity in their undissociated form (i.e., in an acidic environment) due to their ability to penetrate into bacterial cells. In antimicrobial packaging, sorbic acid, benzoic acid, and their salts have found the greatest use [26]. Lactic and acetic acids may be effective in packaging organic acids synthesized by lactic acid bacteria. Calcium alginate films containing these acids limited the growth of *Listeria monocytogenes* and *E. coli* on the surface of beef [83].

Bacteriocins are ribosomally synthesized low molecular weight peptides or proteins that are released extracellularly and have bactericidal or bacteriostatic activity, in particular against a wide range of closely related Gram-positive bacteria, and even against foodborne pathogens. The producing cells are resistant to their own bacteriocins [12,88]. The use of bacteriocins, or lactic acid producing bacteria (LAB), with a wide range of antimicrobial activities can improve safety, control fermentation microflora, accelerate maturation, and extend shelf life, while inhibiting the growth of certain pathogenic bacteria during fermentation, and maturation, generally improving the safety of products [87].

Bacteriocins produced by Gram-positive microorganisms, such as lactic acid bacteria, are preferred because of their preservative properties, especially in the food industry. This preference is also partly due to their broader inhibitory spectrum than Gram-negative microorganisms. The use of bacteriocins in the food industry has increased significantly with increasing concern about the use of chemical preservatives (such as nitrites) that are harmful to human health. Bacteriocins are generally considered safe (GRAS) substances that can be used as food additives or natural preservatives [87,89,90].

Bacteriocins are used in proprietary biocomposites with antimicrobial properties [9,91]; the new possibilities afforded using these in active packaging are still being tested [92,93].

Bacteriocins in purified extract form can be introduced onto the surface of the packaging polymer, or as a starter culture for producing bacteriocins [9,87]. Immobilization of compounds on a carrier placed inside the packaging may be particularly effective and more economical, allowing the controlled release of the substance and a reduction in the amount used, and thus a reduction in cost. The risk of bacteriocin interaction with food ingredients or inactivation of proteolytic enzymes is thus reduced. Various materials, including biodegradable and edible materials, have been considered as potential carriers for bacteriocins or the bacteria producing them; these include silica, corn starch, soy protein, gelatin, calcium alginate, polyethylene, and cellulose. Most often, immobilized bacteriocin is applied to the surface of a food product to protect against the development of harmful microorganisms [92]. This method seems to be beneficial for less-processed foods intended for refrigerated storage. The possibility of preserving fresh fruit and vegetable sprouts using this method is being investigated at the Center for Bioimmobilization and Packaging Material Innovation at West Pomeranian University of Technology in Szczecin, Poland. It has also been shown to be effective with cold-smoked salmon. Studies have reported the effective inhibition of microorganisms such as *Listeria monocytogenes*, *Staphylococcus aureus*, *Bacillus cereus*, and *Alicyclobacillus acidoterrestris* in a package containing bacteriocin, a low cellulose matrix [43].

As in food preservation, the bacteriocin with the greatest usefulness in the production of antimicrobial packaging is nisin, due to its wide spectrum of activity, its long use, and its confirmed safety, being both nontoxic and nonallergenic [94]. Nisin is produced by *Lactococcus lactis* subsp. *lactis*, and has been widely used as a food preservative for over sixty years. The US Food and Drug Administration classifies it as generally recognized as safe (GRAS) and its use as a food preservation additive is permitted in many other countries [9,95]. Nisin is thermally stable and commercially available for use in various food products. This bacteriocin does not lose its antimicrobial activity after pasteurization, sterilization, or other processing methods as it is usually protected by food ingredients. Therefore, this naturally occurring polypeptide has been combined with various biopolymers to produce antimicrobial materials for food preservation [16,96]. Several strategies for integrating nisin in food systems have been used, including nanosize liposomes, emulsions, solid particles, and fibers [95].

Nisin has been proven to effectively inhibit the growth of bacterial pathogens, such as monocytogenes [9,97]. It is often used in dairy products and canned food, effectively inhibiting the growth of heat-resistant spore-forming microorganisms, especially those in the genera *Bacillus*, *Listeria*, and *Clostridium* [9,84]. 

Cartons containing nisin and chitosan have been found to have a positive effect on the microbiological quality of milk and orange juice stored at 10 °C. A low-density polyethylene film with a nisin coating can help to control milk microflora. Nisin also inhibits spore aerobes, lactic acid bacteria, and *Bacillus cereus* in meat products and cheeses protected by LDPE. Wrapped tofu, stored in a refrigerator where nisin was used, was protected against Listeria monocytogenes also after opening of the package. In addition, nisin may be included in edible coatings to prevent the growth of pathogens on the surface of food. It is also worth mentioning that the effects of nisin can be enhanced by the simultaneous use of ethylenediamide tetraacetic acid (EDTA) [84].

Pediocin PA-1 is another bacteriocin used in active packaging [98,99]. This is a bacteriocin of 44 amino acids produced by *Pediococcus acidilactici* [100], whose molecular structure contains four lysine molecules, three histidine molecules, one aspartic acid molecule, and four cysteine molecules connected by disulfide bonds [101]. As an unmodified low molecular weight protein, pediocin shows good thermal stability, stable physicochemical properties, and above all, a broad antimicrobial spectrum. In addition, she has a wide range of adaption to temperature and pH [89,90]. It can be degraded in the human body due to its polypeptide nature and can effectively inhibit various foodborne pathogenic bacteria. Pediocin can thus be developed as a new generation of natural food preservatives [6,102,103], especially given its highly specific inhibitory activity against *Listeria monocytogenes*. Due to its antimicrobial effect, pediocin has many applications in food preservation, which include inoculation of a producing strain or direct addition to food, and more recently, its inclusion in food packaging [104].

Studies by Yin et al. [105] showed the biopreservative efficacy of ACCEL pediocin on chilled fresh fish fillets. Pediocin ACCEL was more effective than nisin on the suppression of *Listeria monocytogenes* growth in refrigerated fish. It is typically known that the occurrence and growth of *L. monocytogenes* in ready-to-eat fish still remains a challenge [105,106,107]. These studies demonstrated that *Pediococcus acidilactici* ALP57, isolated from non-fermented shellfish (oyster, mussels, clams), could synthesize pediocin bac ALP57 (approximately 6.5 kDa, 12,800 AU/mL) that shows the antimicrobial activity against *L. monocytogenes* ESB54 during its exponential growth phase [107,108].

Pediocin can be used to improve food stability, especially in finished meat products, as it shows activity against the growth of *L. monocytogenes* [84,109,110,111], *S. aureus* [112], and *C. perfringens* [113]. This bacteriocin is commercially available under the name ALTA TM 2431 [95,107,109,110,111].

Sanriago-Silva et al. [110] developed a film with a cellulose matrix showing high pediocin-based antimicrobial efficacy that can inhibit the growth of pathogenic bacteria. The researchers stored the ham slices up to 15 days in the prepared matrix and found that 25% and 50% pediocin films had similar antimicrobial activity. The films produce a cycle reduction of 0.5 logs after storing the ham slices for 12 days. Similar properties were demonstrated by the produced nanocomposite films containing nanoparticles of pediocin and ZnO against *L. monocytogens* and *S. aureus* [111]. The composites with the addition of ZnO are of great interest in the research area of food packaging materials due to their improved multifunctional characteristics such as their mechanical, barrier [114,115] and antimicrobial properties [116]. The addition of pediocin results in increased values of elongation at break. On the other hand, the participation of pediocin in nanocomposite films causes its yellowish color. This color is balanced by the addition of ZnO nanoparticles, which ultimately result in a whitish color of the film [117].

Narayanan et al. [118] applied crude pediocin into polyhydroxybutyrate (PHB) to develop antimicrobial films. They show that the film had antimicrobial activity against foodborne pathogens, such as bacteria, and fungi. However, crude pediocin is unable to inhibit fungal growth; only the combined effect of pediocin and PHB films favored antifungal activity [118]. The mechanism of gradual release from the packaging film to the food surface has an advantage over bacteriocin dipping and spraying the foods because recent processing causes the loss the antimicrobial activity of bacteriocins through reaction with food ingredients or their dilution due to migration into the foods [118].

Lacticin is another bacteriocin produced by *L. lactis* subsp. *Lactis.* It is a two-peptide bacteriocin possessing potent activity against Gram-positive bacteria [107]. However, it is worth noting that lacticin differs from nisin in having greater effectiveness and in target specificity In addition, the mechanism of action contrasts with the single nisin peptide as requiring the interaction of two peptides for optimal bactericidal activity [107]. In 2002, Kim et al. [119] showed that lacticin NK24 could slow down the microbial growth on packaged fresh oysters and maintain the chemical quality and extended shelf life significantly [107,119].

Although nisin is the only natural preservative for bacteriocins approved by the Joint Food Additives committee (FAO/WHO), the safety of pediocin has been an indubitable fact. Pediocin is likely the most commercialized natural preservative after nisin [6,112].

The list of bacterocins is quite long; the best known and studied include nisin, pediocin, and lacticin. Table 2 below shows other bacteriocins that are used in food packaging to improve the microbiological safety of packaged food.

### 2.3. Biopolymer Polylactide

One of the goals of the industrial synthesis of lactic acid is to produce a biodegradable polylactide, a polymer of lactic acid [134,135,136].

Polylactic acid (PLA) is considered one of the most promising and environmentally friendly polymers due to its excellent physical and chemical properties, including renewability, biodegradability (100%), and biocompatibility [21,137]. PLA is generally considered safe (GRAS) according to USFDA data. One of the main disadvantages of PLA in food packaging is its high gas and vapor permeability, which limits its use in packaged food with short storage life [46]. As PLA is nontoxic, biocompatible, hydrophilic, water-soluble, and chemically stable, it is usually mixed with other polymers [6,15].

Most lactic acid is produced by fermentation, and the main microorganisms used in this process are lactic acid bacteria. Preference is given to homofermentative species that synthesize minimal amounts of byproducts and to thermophilic species (such as *Streptococcus*), as this allows the process temperature to be raised to about 40–50 °C, limiting the development of unwanted microflora. In addition, strains producing the pure form of l(+) lactic acid, which is more useful in the production of the polymer, are highly valued; the presence of the d(−) isomer in the polymer structure adversely affects the mechanical and rheological properties of polylactide [134,135]. 

Biopolymers are increasingly popular as a replacement for synthetic gasoline polymers. PLA is a biodegradable aliphatic polyester that can be made by fermenting renewable resources, such as corn, cassava, potato, and sugar cane [15,94,138,139,140]. Table 3 below shows the renewable raw materials for the industrial production of lactic acid [94].

Compared with other aliphatic polyesters, PLA has excellent properties, such as mechanical strength, high modulus, biodegradability, biocompatibility, bioabsorbability, transparency, low toxicity, and ease of manufacturing process [22,141,142]. Due to these properties, PLA has many applications, such as agricultural films, biomedical devices, packaging, and the automotive industry [10,143,144]. In addition, it can be used for packaging food in the form of films or coatings [143,144], and it can also be used in antimicrobial systems for packaging food [112,145]. Due to the trend for active and ecological packaging, there is an increasing focus on the use of biomaterials, including cellulose, starch, pectin, and polylalactide [15,146,147]. PLA is a compostable biopolymer that is of interest to the packaging industry due to its rheological and ecological properties [148]. PLA packaging can be produced through many processes, such as film blowing, injection molding, sheet extrusion, blow molding, and thermoforming [95].

Polylactic acid can be made in two ways. The first is the direct polycondensation of lactic acid, which, however, leads to the formation of a polymer with low molecular weight and poorer mechanical properties. A better, although more expensive, method is ring-opening polymerization, which results in a polymer with higher molecular weight [149] and good mechanical properties [104,120].

Polylactide by itself has no known antimicrobial properties, but numerous studies are being carried out to create a polylactide-based packaging with an admixture of various compounds with antimicrobial properties, and in particular antipathogenic properties.

One example is the combination of PLA with the antimicrobial properties of pediocin, which would have many benefits as an active packaging, reducing the risk of foodborne illness for consumers; storage periods could be extended, reducing economic losses; after use, the packaging can be decomposed by composting, reduce the amount of waste produced, and consequently significantly reducing the burden on the environment [104,123,150,151]. One of the safer solutions used in the technology of producing PLA-based packaging may be the use of natural essential oils [152] such as carvacrol [153] and thymol, present in oregano oil [147].

Essential oils are an important group of active compounds in the food industry, serving a wide variety of functions such as flavor, aroma, antimicrobials, antioxidants, and anti-browning agents [152]. Essential oils are usually in the liquid phase under ambient conditions, but they have high volatility that enables them to be used as vapors. Their controlled release in vapors was reported for their mainly antimicrobial and antioxidant properties [152,154]. Due to their physical state, as they can exist in the liquid phase, there are many more methods of stabilizing them than in the case of gaseous compounds [152]. Commonly used methods, such as molecular encapsulation, emulsification, and extrusion, allows to stabilize them and to control their release from various solid matrices. The release from some matrices can occur under ambient conditions and some by external triggers such as high relative humidity and heat, which can free them faster [152,154].

Ahmed et al. [48] have conducted research on the use of additives in the form of ether oils obtained from cinnamon, cloves, and garlic in packaging based on polylactide biopolymers, to determine their antimicrobial effect on the bacterial pathogens *L. monocytogenes* and *S. typhimurium*. The results of their experiment on artificially contaminated samples of cheese packed in foil, showed that the use of cinnamon and clove essential oils as packaging additives reduced some of the amount of pathogens found during storage. Arfat et al. [12] have, in turn, attempted to create an antimicrobial nanopack film consisting of PLA, clove oil, and graphene oxide nanoparticles. They found that the addition of clove oil to the PLA matrix improved the flexibility of the film. The composite film showed excellent antibacterial activity against *Staphylococcus aureus* and *Escherichia coli* [155]. This could be used as an active packaging material that is safe for consumers’ health and contributes to extending the shelf life of the product.

A good example of using PLA to create films with antimicrobial properties is its combination with chitosan [15,142]. PLA also obtains better mechanical properties and better barrier properties. The mechanism of combining these two biocomponents is based on the use of amphiphilic properties of chitosan and hydrophobic properties of PLA [142]. The application of the obtained composite as food packaging material is mainly based on the mechanism of chitosan influence [15]. Its antimicrobial activity in food packaging is based on: Blocking the access to food for the microorganism—the foil is a physical barrier;Blocking the transfer of oxygen, which makes it difficult to access nutrients to the microbial cell;Chelation of nutrients by the chitosan chain, disruption of the functioning of the cell membrane by electrostatic disruption;Death of the microorganism as a result of the dispersion of the chitosan chain inside the cell, which can trigger gene expression or as a result of penetration through the cell nucleus. It can bind DNA, thus inhibiting the replication process, and it can chelate nutrients and metal ions inside the cell [15].

The obtained PLA/chitosan composite film is an interesting alternative to plastics [142,156]. On the one hand, PLA has similar properties to synthetic films, and on the other hand, it is a biodegradable filler similar to the chitosan biopolymer [142]. Plastic films made of, e.g., Polyethylene (PE), Polypropylene (PP), Polyvinyl chloride (PVC), Polycarbonate (PC), or Polyamide (PA) degrade in soil for 150 to 500 years [140,142,156].

The degradability of PLA or a biocomposite with its participation in soil is mainly based on hydrolysis as well as enzymatic and microbiological processes [142,156]. The mechanism of degradation by microorganisms begins only after the hydrolysis of ester bonds under high humidity conditions [140,156]. The biocomposite is decomposed into oligomers, dimers or monomers [142,156]. Degradation in soil with the participation of microorganisms of the *Pseudonocardiaceae* family such as *Kibdelosporangium*, *Amycolatopsis* leads to its mineralization in the form of carbon dioxide, water, or methane [156].

Rizal et al. [142] conducted a study of the biodegradability in soil of pure PLA and PLA/chitosan. The research was carried out for 150 days and the weight loss of the biocomposites was monitored every month. It emerged that pure PLA, due to its hydrophobic properties, is more resistant to microbial attacks, which means that it degrades in the soil much longer than PLA/chitosan. Significant structural disturbance changes for PLA/chitosan were observed after 75 days.

In the case of PLA only, the hydrolysis process itself can take about 60 days (at 60 °C), and last up to 75 days at room temperature [142]. The authors found that the biodegradation rate of the PLA-based biocomposite with the addition of natural polymers was influenced by the amount of their contribution. The acceleration of the degradation process is related to easier access of carbon, e.g., from chitosan or cellulose [142]. They also increased the water-scavenging capacity of PLA (greater hydrophilicity of the compound) and thus facilitated the access of microorganisms [140].

One of the newest solutions seen in the literature is the use of packaging employing “controlled release” technology based on biodegradable polymers, including biopolymers obtained from lactic acid. “Controlled release” is the release of active compounds contained on the surface of the packaging in a controlled manner to extend the shelf life [141,152,154,157,158]. Currently available systems are not sufficient to simultaneously release many active substances in a sustainable manner. For this reason, Biswal and Saha [141] have attempted to develop a packaging technology with double-active antioxidant (fat) and antibacterial properties. The purpose of this type of packaging in one step is to release two different compounds at a sufficient speed that they can inhibit pathogenic bacterial growth in the long-term, as well as inhibit the process of lipid oxidation in food. To achieve this goal, an attempt was made to develop two types of multilayer polymers composed of PLLA (poly-(lactic acid)) and PLGA (poly-(lactic acid-co-glycolic acid) with different viscosities using the emulsion solvent evaporation method. The experiment showed that the use of a set of two compounds as an additive to packaging showed long-lasting antibacterial activity against *Escherichia coli* and *Staphylococcus aureus*, as well as antioxidant activity against lipids for a period of sixty days. These results show for the first time the feasibility of using multilayer microparticles in packaging to extend the shelf life by simultaneously releasing many active substances [141].

PLA accounts for almost 40% of the production of biodegradable plastics from renewable raw materials. PLA is completely broken down and absorbed by microorganisms. It has good barrier properties against oxygen and aroma, resistance to grease, stiffness, strength, and transparency give it great usefulness in the production of food packaging [12,159]. Its disadvantages include low thermal resistance, rather high water vapor permeability, high density compared to polypropylene and polystyrene, and high polarity, which prevents its effective adhesion to nonpolar polymers in multilayer systems [12,104,141]. Polylactide has been subjected to modifications through the addition of various plasticizers, thanks to which its range of application has increased to include bottles, cups, trays, thermoforming films, oriented films, and waste bags [8]. It is also used for laminating paper, producing gardening films, and disposable items (such as cutlery), and is used in agriculture and medicine [120,160]. Polylactide packaging has found application in the dairy industry, with both fresh milk and its products being packed in bottles and cups made of this material; one example is Danone, which launched yogurt in PLA cups in 1998. Polylactide packaging is produced by Nature Works, Huhtamaki, and others, and their products are intended for the storage of desserts, edible oils, and mineral waters [161].

PLA is being utilized as packaging and coating for various food and beverage products such as fruits, vegetables, salads, fresh juices, dairy drinks, yogurts, candies, as well as fish [162].

## 3. Conclusions

Active antimicrobial packaging is an innovative concept in food packaging, which is gaining increasing interest among researchers as well as among producers and representatives of the packaging industry, as they allow the highest quality and safety of food. Antimicrobials embedded in or lined with packaging materials have significant inhibitory activity against various types of microorganisms that contribute to food spoilage. An example of the growing interest in chitosan-based packaging.

The centuries-old use of lactic acid bacteria for preserving food in the fermentation process has not lost its relevance. The gradual development of science and technology has allowed a thorough understanding of these specific microorganisms, allowing their use to be expanded. Lactic fermentation bacteria inhibit the growth of pathogenic and spoilage microorganisms. They produce a number of metabolites that inhibit the growth of Gram-positive and Gram-negative bacteria, as well as molds and yeast. Some lactobacilli metabolites have also found application in the production of food packaging. Bacteriocins and organic acids are often used in active packaging, where they prevent spoilage of food products, thus contributing to improving the quality, nutritional value, sensory aspects, and safety of manufactured food. Lactic acid synthesized on an industrial scale is also used to produce packaging biopolymer, which is gaining increasing importance and use, contributing to a significant improvement in environmental conditions.

## Figures and Tables

**Figure 1 molecules-26-03735-f001:**
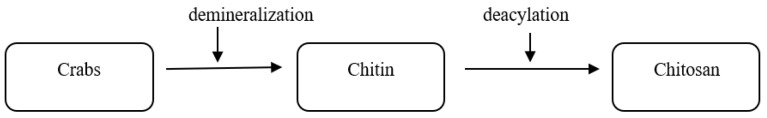
Preparation of chitosan.

**Table 1 molecules-26-03735-t001:** Chitosan-based antimicrobial packaging and its potential applications.

Product Preserved	Packaging Material	Antimicrobial Agent	Reference
Cucumber	Chitosan	Limonene	[69,70]
Cucumber	Chitosan/carnauba wax	Oregano essential oil (OEO)	[71]
Tomato	Chitosan	TiO_2_ nanoparticles	[72]
Strawberries	Chitosan/CMC	Chitosan	[73]
Mushroom	Chitosan	Galic acid	[74]
Rainbow trout fillet	Chitosan	Grape seed extract	[15,75]
Shrimps	Chitosan	Carvacrol	[76]
Chicken	Chitosan	Acerola residue extract	[77]
Poultry	Chitosan	Ginger oil	[78]
Chicken	Chitosan/PET	Plantaricin	[62]
Lamb meat	Chitosan	Satureja plant oil	[15,79]
Ham	Chitosan/starch	Gallic acid	[80]
Cheese	Chitosan/PVA	TiO_2_	[81]

**Table 2 molecules-26-03735-t002:** Application of bacteriocin from LAB in food preservation.

Types of Bacteriocin	Producing Strain	Food Application	Targeted Pathogens	References
Nisin	*L. lactis* spp.	Cheddar cheese	*L. monocytogenes, S. aureus*	[112,113]
		Milk and milk products	*B. cereus, C. botulinum* and *C. perfringens*	[113,120,121]
		Meat and sausages	*C. botulinum* and *L. monocytogenes*	[122]
Pediocin	*P. acidilactici*	Dried sausages and fermented meat products	*L. monocytogenes* and *C. perfringens*	[113,123]
		Fresh beef, vacuum-packed beef, cottage cheese, ice cream mix	*Ln. mesenteroides*	[113,124]
		fish fillets, chicken meat	*L. monocytogenes*	[113,125]
		Sous vide products	*B. subtilis*, *B. licheniformis*	[126]
Lacticin	*L. lactis* spp.	Milk and milk products	A medium spectrum of *C. tyrobutyricum* and *L. monocytogenes*	[113,127,128]
Sakacin	*L. sakei*	Meat product	*L. monocytogenes*	[129]
Enterocin AS-48	*E. faecalis A-48-32*	Non-fat hard cheese	*B. cereus*	[113,130]
		Fruit juices	*A. aciditerrestris*	[113]
		Apple cider	*B. licheniformis*	[113]
		Vegetable soups and purre	*B. cereus, Paenibacillus spp., B. macroides*	[113,131]
		Cooked ham	*L. monocytogenes*	[113]
		Skimmed milk and non-fat unripened soft cheese	*B. cereus*	[112,132]
Enterocin A	*L. lactis MG1614*	Cottage cheese	*L. monocytogenes*	[133]
Bacteriocin 7293	*W. hellenica BCC 7239*	Meat and meat products	*P. aeruginosa, E. coli,* and *S. typhimurium*	[122]

**Table 3 molecules-26-03735-t003:** Renewable raw materials used in the industrial production of lactic acid.

Renewable Raw Materials		
Starch Raw Materials	Cellulosic Raw Materials and Hemicellulosic Raw Materials	Industrial Waste Products
potatoes wheat maize rice rye oat barley sorghum	straw of rice, wheat, maize lucerne fibers waste wood waste paper	molasses whey

## Data Availability

Data sharing not applicable.

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
