# Peer review of "Antimicrobial Food Packaging with Biodegradable Polymers and Bacteriocins"

_molecules, 2021, doi:10.3390/molecules26123735_

Round 1
Reviewer 1 Report
Dear Editor,
I was invited to evaluate the review « Antimicrobial Food Packaging with Biodegradable Polymers and Bacteriocins» by a Gumienna and Górna.
In their review, the authors summarized literature data regarding the use of antimicrobial polymers and bacteriocins in food and drink packaging. As mentioned by the authors, antimicrobial and active packaging are at the forefront of current research and development for food packaging (and not only). This includes the use of natural polymers with antimicrobial properties such as chitin derivatives. Thus, chitosan films have the potential to be used as a packaging material for food to prevent microbial contamination through activity against a wide range of pathogenic and food spoilage microbes. Polylactic acid (PLA) is another exemple cited by the authors with very good physical and chemical properties including renewable, biodegradability, biocompatibility, and its safety. The aim of the authors in this review is to summarize all existing possibilities of using various components in combination with chitosan, PLA or bacteriocins in order to make antimicrobial food packaging able to improve/increase the quality, delay the spoilage of packaged food, as well as increase the safety and shelf life of food in addition to be eco-friendly.
I found this review very interesting and informative. It is well documented and bring a nice overview of the field.
I have however few minor comments :
1- It will be good for the reader to have the information about how chitosan of PLA act as antimicrobial agent alone (what is/are their mechanism of action ?) Please ad dit clearly into the text.
2- Please also add informations regarding the half-life of those antimicrobial films done with chitosan or with PLA compared to plastic films.
3- Regarding the film with added antimicrobial agents, do the authors have data concerning their degradability ? One can expect the antimicrobial film to degrade slower (or not at all) due to the fact they kill micro-organisms able to degrade them…
regards
Author Response
Poznan, Jun 14th, 2021
Response to Reviewer #1
Thank you very much for your time and review of the publication “Antimicrobial Food Packaging with Biodegradable Polymers and Bacteriocins” by a Gumienna and Górna.
The suggestions made allowed to improve the manuscript and contributed to its better understanding. Below, I present detailed corrections made in accordance with the reviewer's suggestions.
Comment 1: It will be good for the reader to have the information about how chitosan of PLA act as antimicrobial agent alone (what is/are their mechanism of action ?) Please ad dit clearly into the text.
- Response: in body text was added a new lines 486 - 501, and new reference
A good example of using PLA to create films with antimicrobial properties is its combination with chitosan [15,142]. PLA also obtains better mechanical properties and better barrier properties. The mechanism of combining these two biocomponents is based on the use of amphiphilic properties of chitosan and hydrophobic properties of PLA [142]. The use of the obtained composite as food packaging is mainly based on the mechanism of chitosan influence [15]. Its antimicrobial activity in food packaging is based on:
- blocking the access to food for the microorganism - the foil is a physical barrier,
- blocking the transfer of oxygen, which makes it difficult to breathe and access nutrients to the microbial cell,
- chelation of nutrients by the chitosan chain, disruption of the functioning of the cell membrane by electrostatic disruption,
- death of the microorganism as a result of the dispersion of the chitosan chain inside the cell, which can trigger gene expression or as a result of penetration through the cell nucleus, it can bind DNA, thus inhibiting the replication process, and it can chelate nutrients and metal ions inside the cell [15].
Comment 2: Please also add informations regarding the half-life of those antimicrobial films done with chitosan or with PLA compared to plastic films.
- Response: Commentary 2 was answered together with comment 3 combining the description of the possibility of decay of plastic films with the degradation of biopolymers and their mechanism. No detailed data on the half-life of antimicrobial films or plastic films were found, therefore the possibility of degradation of these films was included on the example of PLA, PLA /chitosan. In the case of plastic foils, data on their decomposition time have been found in the literature.
Comment 3: Regarding the film with added antimicrobial agents, do the authors have data concerning their degradability ? One can expect the antimicrobial film to degrade slower (or not at all) due to the fact they kill micro-organisms able to degrade them…
- Response: in body text was added a new lines 502 - 526, and new reference
The obtained PLA / chitosan composite film is an interesting alternative to plastics [142,156]. On the one hand, PLA has similar properties to synthetic films, on the other hand, it is a biodegradable filler, similar to the chitosan biopolymer [142]. Plastic films made of e.g. Polyethylene (PE), Polypropylene (PP), Polyvinyl chloride (PVC), Polycarbonate (PC) or Polyamide (PA) degrade in soil for 150 to 500 years [140,142,156].
The degradability of PLA or a biocomposite with its participation in soil is mainly based on hydrolysis as well as enzymatic and microbiological processes [142,156]. The mechanism of degradation by microorganisms begins only after the hydrolysis of ester bonds under high humidity conditions [140,156]. The biocomposite is decomposed into oligomers, dimers or monomers [142,156]. Degradation in soil with the participation of microorganisms of the Pseudonocardiaceae family such as Kibdelosporangium, Amycolatopsis leads to its mineralization to the form of carbon dioxide, water or methane [156].
Rizal et al. [142] conducted a study of the biodegradability in soil of pure PLA and PLA / chitosan. It turned out that pure PLA, due to its hydrophobic properties, is more resistant to microbial attacks, which means that it degrades in the soil much longer than PLA / chitosan. Significant structural disturbance changes for PLA / chitosan were observed after 75 days.
In the case of PLA only, the hydrolysis process itself can take about 60 days (at 60 C), and last up to 75 days at room temperature [142]. The authors found that the biodegradation rate of the PLA-based biocomposite with the addition of natural polymers is influenced by the amount of their addition. The acceleration of the degradation process is related to easier access of carbon, eg from chitosan or cellulose [142]. They also increase the water-scavenging capacity of PLA (greater hydrophilicity of the compound) and thus facilitate the access of microorganisms [140].
- new reference:
- Brebu, M. Environmental Degradation of Plastic Composites with Natural Fillers—A Review. Polymers2020, 12, 166. doi.org/10.3390/polym12010166
- Zaaba, N.F.; Jaafar, M. A review on degradation mechanisms of polylactic acid: Hydrolytic, photodegradative, microbial, and enzymatic degradation. Polym. Eng. Sci.2020, 60, 2061–2075. doi.org/10.1002/pen.25511
- Response: English language was corrected also by native speaker.
Thank you for the work put to improve our paper.

Reviewer 2 Report
Here are some improvements that authors can make to make it easier to understand.
In the introduction nothing is said about chitosan, I believe that the wording of the text can be improved if the chitosan mentioned in the reference is mentioned together with the PLA [13]
Line 47: It says: "(PLA), which is particularly useful in food packaging [10,12,13]". It should say: (PLA) and chitosan, which are particularly useful in food packaging [10,12,13]
Line 91: insert any example or reference
Line 92 insert any example or reference
Lines 181 to 193: For the reader to understand why gallic acid is mentioned between lines 181 to 188, I suggest that authors reorganize this paragraph or section 2.1 by introducing phenolic compounds and their adducts or complexes conjugated with chitosan.
At the end of section 2.1 it would be desirable to mention the existence of antimicrobial films of chitosane, polyphenols and nanosilver (if considered appropriate), and if there is synergistic behavior, as mentioned later at the beginning of section 2.2 for Lactic acid.
Author Response
Poznan, Jun 14th, 2021
Response to Reviewer #2
Thank you very much for your time and review of the publication “Antimicrobial Food Packaging with Biodegradable Polymers and Bacteriocins” by a Gumienna and Górna.
The suggestions made allowed to improve the manuscript and contributed to its better understanding. Below, I present detailed corrections made in accordance with the reviewer's suggestions.
Comment 1: In the introduction nothing is said about chitosan, I believe that the wording of the text can be improved if the chitosan mentioned in the reference is mentioned together with the PLA [13]
- Response: in body text was added a new lines 50- 52, and new reference
- Amankwaah, C.; Li, J.; Lee, J.; Pascall, M.A. Antimicrobial Activity of Chitosan-Based Films Enriched with Green Tea Extracts on Murine Norovirus, Escherichia coli , and Listeria innocua. Int. J. Food Sci. 2020, 2020, 1–9, doi:10.1155/2020/3941924.
Comment 2 : Line 47: It says: "(PLA), which is particularly useful in food packaging [10,12,13]". It should say: (PLA) and chitosan, which are particularly useful in food packaging [10,12,13]
- was corrected, now line 49
Comment 3: Line 91: insert any example or reference
- was added 3 reference, now line 102
11.Malhotra, B.; Keshwani, A.; Kharkwal, H. Antimicrobial Food Packaging: Potential and Pitfalls. Front. Microbiol. 2015, 6, doi:10.3389/fmicb.2015.00611
22.Cotter, P.D.; Hill, C.; Ross, R.P. Bacteriocins: Developing Innate Immunity for Food. Nat. Rev. Microbiol. 2005, 3, 777–788, doi:10.1038/nrmicro1273.
- Fang, Z.; Zhao, Y.; Warner, R.D.; Johnson, S.K. Active and Intelligent Packaging in Meat Industry. Trends Food Sci. Technol. 2017, 61, 60–71, doi:10.1016/j.tifs.2017.01.002.
Comment 4: Line 92 insert any example or reference
- was added 2 reference, now line 104
- Malhotra, B.; Keshwani, A.; Kharkwal, H. Antimicrobial Food Packaging: Potential and Pitfalls. Front. Microbiol. 2015, 6, doi:10.3389/fmicb.2015.00611
23. Khan, I.; Oh, D.-H. Integration of Nisin into Nanoparticles for Application in foods. Innov. Food Sci. Emerg. Technol. 2016, 34, 376–384, doi:10.1016/j.ifset.2015.12.013.
Comment 5: Lines 181 to 193: For the reader to understand why gallic acid is mentioned between lines 181 to 188, I suggest that authors reorganize this paragraph or section 2.1 by introducing phenolic compounds and their adducts or complexes conjugated with chitosan.
- Response: in body text was added a new lines 199- 209, and new reference
The compounds contained in the discussed extracts with strong antimicrobial properties are the already mentioned polyphenol compounds [48–50]. They belong to the natural antioxidants formed on the acetate-malonic pathway from carbohydrates. Their main source are plants, eg fruit and vegetables [49,56]. Depending on the structure, there are: derivatives of hydroxybenzoic acid (gallic acid, ellagic acid), derivatives of hydroxycinnamic acid (p-coumaric, coffee, ferulic and chloragenic acid) and stilbenes (e.g. resveratrol), tannins and flavonoids including such subgroups as: flavonols (e.g. quercetin, campferol, myretin, morin, flavones (luteolin, apigenin), flavanones, flavanols, isoflavones (daidzein, genistein, glycitein), catechins and anthocyanidins (cyanidine, delphinidin, malvidin, pelargonidine). Their common trait is the presence of phenolic groups in the molecules [56].
- Ballard, C.R.; Maróstica, M.R. Health Benefits of Flavonoids. In Bioactive Compounds; Elsevier, 2019; pp. 185–201.
Comment 6: At the end of section 2.1 it would be desirable to mention the existence of antimicrobial films of chitosane, polyphenols and nanosilver (if considered appropriate), and if there is synergistic behavior, as mentioned later at the beginning of section 2.2 for Lactic acid.
- Response: in body text was added a new lines 242- 252, and new reference
Another interesting solution is the combination of antimicrobial chitosan, tea polyphenols and silver nanoparticles (AgNPs) [65]. A novel one-pot method was used to create this nanocomposite film. The addition of tea polyphenols (TP) was used here, adding them to chitosan as a reducing agent for AgNPs, but above all as a binding agent and a good antioxidant. Already the work carried out by Thomas et al. [66] showed high antimicrobial effectiveness of films made of chitosan and silver nanoparticles against Escherich coli and Bacillus bacteria. These films were obtained by synthesizing a silver particle with chitosan by reducing silver ions in an acid solution of AgNO3 [66].
On the other hand, the introduction of TP to chitosan thogeder with AgNPs resulted in an improvement in the mechanical properties of the obtained composite, as well as in a higher antioxidant and antimicrobial activity [65].
- new reference
- Zhang, W.; Jiang, W. Antioxidant and antibacterial chitosan film with tea polyphenols-mediated green synthesis silver nanoparticle via a novel one-pot method. Int. J. Biol. Macromol. 2020, 155, 1252–1261, doi:10.1016/j.ijbiomac.2019.11.093.
- Thomas, V.; Yallapu, M.M.; Sreedhar, B.; Bajpai, S.K. Fabrication, Characterization of Chitosan/Nanosilver Film and Its Potential Antibacterial Application. J. Biomater. Sci. Polym. Ed. 2009, 20, 2129–2144, doi:10.1163/156856209X410102.
- and was corrected line: 212-216 it was: The publicly available literature indicates the promising potential of gallic acid in the development of antimicrobial packaging materials to reduce the effects of pathogens and bacteria that contribute to food spoilage [49,50,63,65–67]. As an additive to packaging, GA increases flexibility, thus acting as a plasticizer and eliminating the classic problem of brittleness. It can thus be used in the production of chitosan films for food packaging [65,68]. Now is line 253-257: Thus, the publicly available literature indicates the promising potential of polyphenolic compounds in the development of antimicrobial packaging materials to reduce the effects of pathogens and bacteria that contribute to food spoilage [49,50,63,65–67]. As packaging additives, they also improve the mechanical properties of the obtained chitosan-based composites [65,68].
Thank you for the work put to improve our paper.

Round 2
Reviewer 2 Report
The authors have made all the changes we suggested and have greatly improved the article. Therefore and from my point of view I consider it adequate to be published.